# A Global Review of Publicly Available Datasets Containing Fundus Images: Characteristics, Barriers to Access, Usability, and Generalizability

**DOI:** 10.3390/jcm12103587

**Published:** 2023-05-21

**Authors:** Tomasz Krzywicki, Piotr Brona, Agnieszka M. Zbrzezny, Andrzej E. Grzybowski

**Affiliations:** 1Faculty of Mathematics and Computer Science, University of Warmia and Mazury, 10-710 Olsztyn, Poland; tomasz.krzywicki@matman.uwm.edu.pl; 2Department of Ophthalmology, Poznan City Hospital, 61-285 Poznań, Poland; 3Faculty of Design, SWPS University of Social Sciences and Humanities, Chodakowska 19/31, 03-815 Warsaw, Poland; 4Institute for Research in Ophthalmology, Foundation for Ophthalmology Development, 60-836 Poznań, Poland

**Keywords:** color retinal fundus images, health data, image datasets, ophthalmic diseases

## Abstract

This article provides a comprehensive and up-to-date overview of the repositories that contain color fundus images. We analyzed them regarding availability and legality, presented the datasets’ characteristics, and identified labeled and unlabeled image sets. This study aimed to complete all publicly available color fundus image datasets to create a central catalog of available color fundus image datasets.

## 1. Introduction

Research and healthcare delivery are changing in the digital age. Digital health research and deep-learning-based applications are promising to transform some of the ways we care for our patients and expand access to healthcare in both developed and underprivileged regions of the world [1,2,3]. Automated screening for diabetic retinopathy (DR) is one facet of this transformation, with deep-learning algorithms already supplementing clinical practice in different parts of the world [4,5].

As the barrier to entry for creating deep-learning-based applications significantly diminished over the last few years, many smaller companies and institutions now attempt to create their algorithms for healthcare, particularly for image-based analysis [6]. Radiology and ophthalmology are medical specialties for which deep learning is most applicable due to their reliance on images and visual analysis [7,8,9]. Color fundus photos, retinal and anterior chamber optical coherence tomography (OCT) scans, and visual field analyzer reports can all lend themselves to automatic analysis for various possible pathologies [9,10].

Creating these algorithms requires sizable numbers of initial images, both with and without pathology. Data are needed in every step of developing a deep-learning application. In the modern world, with electronic healthcare records, centralized imaging storage, and the pervasiveness of digital solutions and storage, such data are generated worldwide in massive quantities due to access, cost, and healthcare issues. Privacy and data governing laws, lack of centralized databases, heterogeneity within particular datasets, lack of or insufficient labeling, or the sheer volume of images required to access such data are often challenging. This article provides an overview of the repositories containing color fundus photos. We analyze them in terms of availability and legality of use. We present the characteristics of datasets and identify labeled and not labeled sets of images. We also analyze the origin of the datasets. This review aims to complement all publicly available color fundus image datasets to create a central catalog of what is currently available. We list the source of each dataset, their availability, and a summary of the populations represented.

Khan and colleagues have previously identified, described, listed, and jointly reviewed 94 ophthalmological imaging datasets [11]. However, this review is outdated now. We aim to provide an update on those datasets’ current state and accessibility.

Khan et al.’s [11] work includes 54 repositories of color fundus photos. At the time of writing this article, only 47 were available. In this work, we have added 73 repositories of color fundus photos, samples of the content of each of them (https://shorturl.at/hmyz3; accessed on 16 May 2023), a tool for automatic content inspection of current or future repositories, accurate access type information (form/registration/e-mail to the authors/no difficulties), degree of difficulty of access to the repository, total file size, image sizes, photo descriptions, information about additional artifacts and legality of use, along with any required papers to be cited.

In addition, we classified the datasets into the following five categories:The availability of the datasets;A breakdown of the legality of using the datasets;A classification of the image descriptions;Geographical distribution of the datasets that are available by continent and country.

## 2. Methods

We used the information presented in the review of publicly available ophthalmological imaging datasets [11]. Each dataset includes details about its accessibility, data access, file types, countries of origin, number of patients undergoing examination, number of all images taken, ocular diseases, types of eye examinations performed, and the device used. We have extended the information about all the datasets marked as available in the review mentioned above. We found 47 such color fundus image repositories.

We then used well-known tools to find other repositories not described in the mentioned papers. Searching for color fundus image repositories consisted of typing different types of terms into three types of search engines, including “fundus”, “retina” and “retinal image” along with the words “dataset”, “database” and “repositories”. The exact search was done in the Google search engine and the Google Dataset search engine, designed to search online datasets. Google Dataset Search is designed for online repository discovery and supports searching for tabular, graphical, and text datasets. Indexing is available for publishing their dataset with a metadata reference schema. All results from the search describe the dataset’s contents, direct links, and file format. Google searches also included terms related to images of the retina and terms related to datasets. For both searches, we considered the first ten pages of results. We found 17 unique repositories: 5 using the Google search engine and 12 using the Google Dataset search engine.

The third search engine we chose was Kaggle. Kaggle is a data science and artificial intelligence platform on which users can share their datasets and examine the datasets shared by others. Kaggle datasets are open-sourced, but to determine for what purposes these datasets can be used, we need to check the datasets’ licenses. The vast majority of Kaggle datasets are reliable. We can judge a dataset’s reliability by looking at its upvotes or reviewing the notebooks shared using the dataset. We used the same types of terms as with the Google search engine and Google Dataset search engine. We found 61 unique repositories. We investigated the actual condition of files with their total size, image sizes, information about image description, additional artifacts found in images, issues of legality of data used in scientific applications, and visualizations of sample data. We did not exclude any color fundus image datasets based on the age, sex, or ethnicity of the patients from whom data was collected. We also included datasets of all languages and geographic origins.

### 2.1. Dataset Checking Strategy

We noticed that the levels of access to the datasets varied: from fully accessible to available on request after sending a request to the authors. Some datasets were also unavailable. In this article, we have defined access levels as follows:(1)Fully open;(2)Available after completing a form;(3)Available after account registration (and possible approval by the authors);(4)Available after sending an email to authors and approval from them;(5)Not available.

After accessing, we manually checked each dataset described in 1 by downloading them to extract information about file status, sizes, and additional artifacts found in the images. Most available datasets were available as compressed files (ZIP/RAR/7Z), but some were available as separate files. We determined the sizes of the datasets in which the files were delivered separately by downloading all files and summing their sizes. We have prepared a tool to automatically generate the discussed information on repository contents and image samples—Ophthalmic Repository Sample Generator (Section 4.1). We also manually checked the content of each randomly selected image to see if there were any additional artifacts.

### 2.2. Image Descriptions and Legality of Use

All the datasets we reviewed were described on dedicated web pages or in scientific publications. Based on these sources, we have determined methods of describing images included in these datasets.

We noticed the following types of image descriptions:(1)Manually assigned labels corresponding to diagnosed ocular diseases, image quality, or described areas of interest;(2)Manual annotations on images indicating areas of interest;(3)No descriptions.

We have also extracted information on the legality of data use from websites dedicated to the datasets. We noticed the following approaches to defining the legality of the use of data contained in the datasets:(1)Notifying the authors of the datasets of the results and awaiting permission to publish the results;(2)References to the indicated articles or the dataset in the case of publication of the results;(3)No restrictions.

## 3. Data Availability

In the analysis, we used 121 publicly available datasets containing color fundus images [12,13,14,15,16,17,18,19,20,21,22,23,24,25,26,27,28,29,30,31,32,33,34,35,36,37,38,39,40,41,42,43,44,45,46,47,48,49,50,51,52,53,54,55,56,57,58,59,60,61,62,63,64,65,66,67,68,69,70,71,72,73,74,75,76,77,78,79,80,81,82,83,84,85,86,87,88,89,90,91,92,93,94,95,96,97,98,99,100,101,102,103,104,105,106,107,108,109,110,111,112,113,114,115,116,117,118,119,120,121,122,123,124,125,126,127].

## 4. Code Availability

### 4.1. Ophthalmic Repository Sample Generator

We developed a generator of pseudo-random samples from publicly available repositories containing color fundus photos. The generator is written in Python 3 programming language and is available on GitHub (https://github.com/betacord/OphthalmicRepositorySampleGenerator; accessed on 16 May 2023).

The prepared tool facilitates the manual inspection of the contents of repositories. The program obtains the URL of a given repository (or ID on the Kaggle platform) and the sample size (n). The operation result will be a pseudo-random selection of n color fundus photos from the repository and a CSV file containing extracted attributes representing the entire repository. The tool can be easily run on a local computer or in a cloud environment. The general scheme of the generator is shown in Figure 1.

As an input, the generator takes the sample size, the repository URL, the data output file path, the temporary full data path, the repository sample output path, the repository type, and the output CSV file path.

The sample size is an integer number representing the size of the random output sample of photos from the repository.A repository URL is a string representing a direct URL of the image file; e.g., for the Kaggle dataset, the schema is [username]/[dataset_id]. In the case of a Kaggle competition, it is ID.The data output file path is a string representing the output file with the downloaded repository content.The temporary full data path is a string representing the temporary path to which the repository will be extracted.The repository sample output path is a string representing the path where a randomly selected repository sample will be placed.The repository type is an integer representing the type of the repository source: 0 for classic URL, 1 for Kaggle competition, and 2 for Kaggle dataset.The output CSV file path is a string representing the path where the CSV file will be saved (separated by;) containing information about the repository.

Therefore, external parameters characterizing the size of the generated sample of images, data source, temporary paths, source type, and paths to the output files should also be included in the tool’s run.

## 5. Results

In Table 1, we included the results of our review. In total, we checked 127 repositories containing color fundus images, of which 120 were currently available, and seven were unavailable due to a non-existent URL. Downloading one dataset was prevented due to a critical server error, and one dataset was delivered as a corrupt zipped file. We have described the characteristics only for the available datasets. We also generated a sample of their content and placed it in the cloud (https://shorturl.at/hmyz3; accessed on 16 May 2023).

### 5.1. Data Access

Out of the 127 available datasets, we marked 37 as fully open, 6 as available after completing a form, 75 as available after account registration, 2 as available after sending an email to authors and approval from them, and 7 as not available, as can be seen in Figure 2.

### 5.2. Characteristics of Datasets

Almost all (122 out of 124) of the datasets could be downloaded as zipped files, and only 2 could be downloaded separately. There was a problem with the extension on three of the zipped files that contained datasets. In 59 datasets, all images had the exact dimensions (in pixels), but there were 68 unique ones. Images from nine different datasets contained additional artifacts such as dates, numbers, digits, color scales, markers, icons, arteries, vessels, veins, and key points marked on the photographs.

### 5.3. The Legality of Use and Image Descriptions

Out of the 127 datasets, the authors of 3 of them provided a note about the need to inform them about the obtained results. Authors of 44 datasets provided information about the need to cite the indicated works using the provided data and publishing the results. Over two-thirds, or 80 datasets, had no restrictions on use. Figure 3 shows the full breakdown of the legality of using data contained in the datasets.

Eighty-nine datasets were labeled with the images assigned to them. Thirty datasets had areas of interest labeled on the images. Sixteen datasets did not have any descriptions. A full breakdown of the image descriptions is shown in Figure 4.

## 6. Discussion

Publicly available datasets remain important in digital health research and innovation in ophthalmology. Although, on the whole, the number of publicly available color fundus image datasets is growing, it is an ongoing process with new datasets arriving and older datasets becoming inaccessible. A central repository or listing for ophthalmic datasets, coupled with the low discoverability of many of the datasets, constitutes a significant barrier to access to high-quality representative data suitable to a given purpose. However, this article provided an up-to-date review and discussion of available color fundus image datasets.

Health data poverty, in this case, scarcity of color fundus image datasets originating from underprivileged regions, particularly Africa, is cause for concern. Although the relationship between patients’ ethnicity, background, and other attributes and fundus features is not clearly documented, the lack of representative datasets might lead to ethnic or geographical bias and poor generalizability in deep-learning applications. The recent relative lack of datasets might mean underrepresented regions miss out on future data-driven screening and healthcare solutions benefits.

The study published by Khan and colleagues was the first comprehensive and systematic listing of public ophthalmological imaging databases. In their work, out of 121 datasets identified through various searching strategies, only 94 were deemed truly available, with 27 databases being inaccessible even after multiple attempts spaced weeks apart [11]. Therefore roughly one-fourth of datasets were inaccessible at the time of their review in work mentioned above. It is in line with our findings in writing this update. Out of 94 datasets marked available by the authors, only 74 were available at the time of preparing this article, just over 1 year from the initial paper publication of Khan and colleagues and just 15 months after the first online publishing. Therefore, access to over one-fifth (21%) of datasets was lost in fewer than 2 years, similar to the 22% found inaccessible in the original review. Almost all of the datasets that became unavailable since the publishing of Khan’s study were offline—the dedicated websites are unreachable, with two unavailable due to errors.

Of the unavailable datasets, 7 became unavailable during the identification, verification, and review of the newly discovered datasets out of 127 color fundus image repositories initially identified for this analysis. Although the initial period between publishing the individual datasets and becoming inaccessible has yet to be discovered, it is clear that datasets going offline or otherwise becoming unreachable is an ongoing process, and information on availability can quickly become obsolete. It is important to note that while Khan et al. published a list of datasets containing OCT and other imaging modalities, this review focuses specifically on datasets of color fundus images [11].

Given adequate citation of sources, all but three of the datasets allowed unrestricted access and publishing of results for scientific, non-commercial purposes. The two exceptions required approval from dataset authors before publishing any results, which may limit their usability in scientific regard, leaving potential publication opportunities to the whim of original dataset authors. More than half of datasets do not impose any restrictions and do not explicitly require citations, though quoting sources is one of the fundamental ethical principles of scientific use.

Most datasets contain additional information about individual images. More than half of the datasets (66%) contained manually assigned text-based labels corresponding to diagnosed ocular diseases, image quality, or described areas of interest. One-fifth (22%) of datasets contained annotations indicating areas of interest or pathology. Only 12% of datasets contained raw images without metadata for individual images.

Figure 5 and Figure 6 explore the geographical origins of the datasets.

Almost half of the datasets for which a region of origin could be established originated from Asia, with Europe making up another one-third. Overall, out of 73 datasets, 24 originated from outside of Europe or Asia, with none of the datasets originating from Africa and a nearly equal split between North and South America. The distribution of datasets available from individual countries is shown in Figure 6. Although dataset origin relates to the location of the person or organization sharing the dataset and does not necessarily represent the origin of patients’ images, the complete lack of images from Africa is concerning.

Africa, particularly sub-Saharan Africa, is a vastly underserved region with one of the lowest numbers of ophthalmologists in the population globally [167]. There are, on average, three ophthalmologists per million populations in sub-Saharan Africa, compared to about 80 in developed countries [167]. Although this is likely one of the reasons for the lack of available datasets from the region, it is also the rationale for the need for datasets from this region. Digital healthcare solutions, including deep-learning software, may help alleviate some of the healthcare disparities in the region. However, these require development or at least validation on the target validation to avoid any potential for racial or other population-specific trait bias [168]. It remains steadfast in the case of color fundus images where other than background fundus pigmentation, the influence of patients’ attributes such as age, sex, or race on fundus features and their variations are not well known. It is also currently being tackled using deep-learning methods [169]. The suspicion of poor generalizability in populations outside of the ethnic or geographical scope of the initial training image data and, subsequently, the need for the development and validation of multi-ethnic populations is not a new concept in the automated analysis of color fundus images [11,149,170,171]. Serener et al. have shown that the performance of deep-learning algorithms for detecting diabetic retinopathy in color fundus images varies based on geographical or ethnic traits of the training and validation populations [171].

## 7. Conclusions

Open datasets are still crucial for digital health research and innovation in ophthalmology. Even though the public has access to more datasets with color fundus images, new datasets are always being added, making older datasets inaccessible. There are only a few places to store or list ophthalmic datasets, and many of them are hard to find, making it hard to obtain high-quality, representative data useful for a given purpose. This paper discussed the many color fundus image datasets that are now available and gave an up-to-date review.

## Figures and Tables

**Figure 1 jcm-12-03587-f001:**
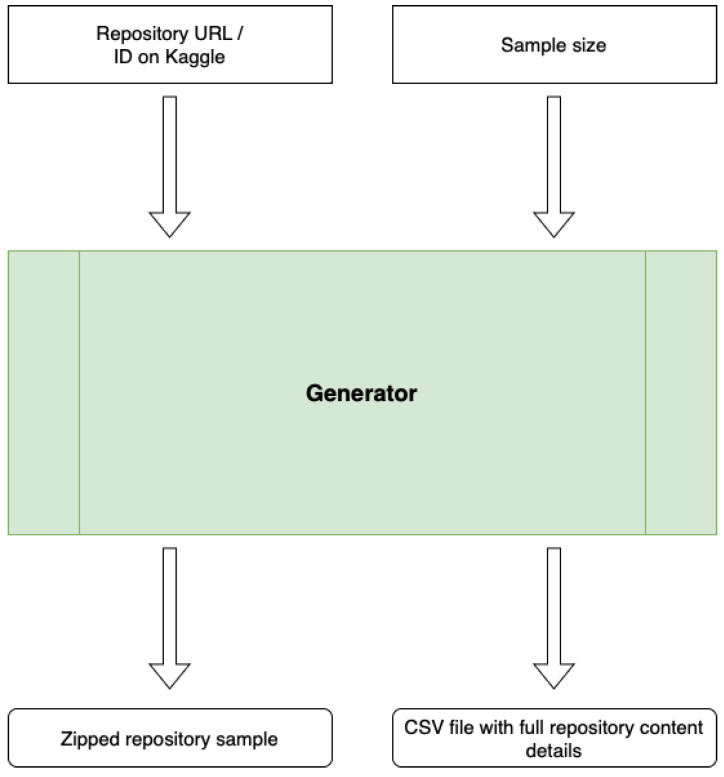
General scheme of the Ophthalmic Repository Sample Generator.

**Figure 2 jcm-12-03587-f002:**
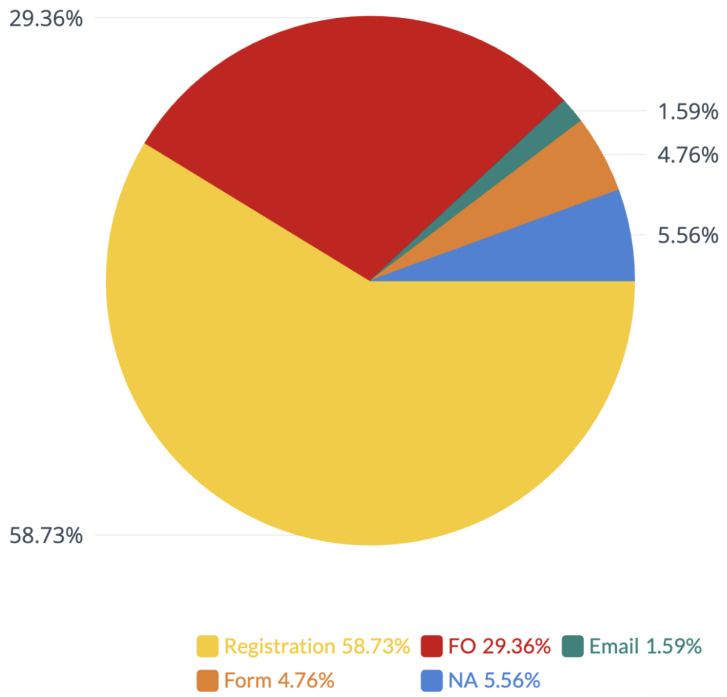
Availability of datasets. FO = fully open, FORM = available after completing a form, Registration = available after account registration (and possible approval by the authors), Email = available after sending an email to authors and approval from them, NA = not available.

**Figure 3 jcm-12-03587-f003:**
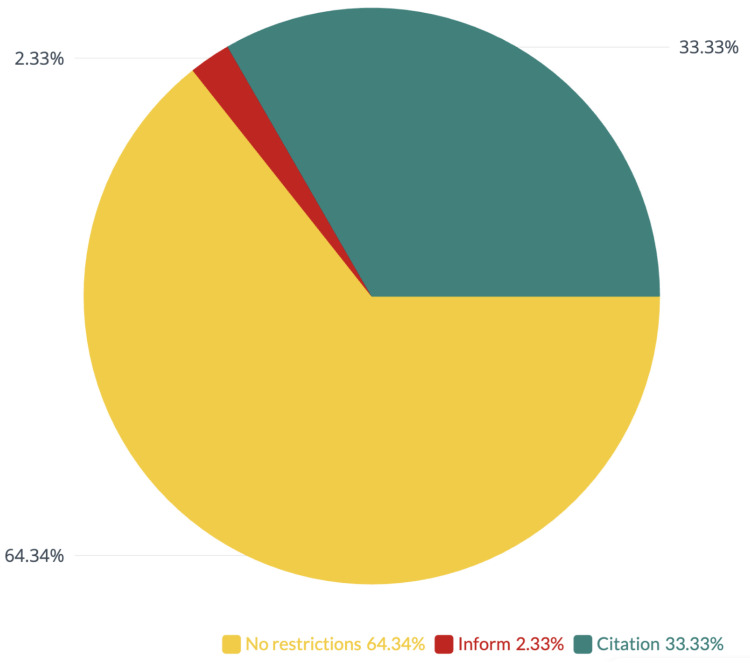
Breakdown of the legality of using data. Inform = notifying the authors of the datasets of the results and awaiting permission to publish the results, Citation = references to the indicated articles in the case of publication of the results.

**Figure 4 jcm-12-03587-f004:**
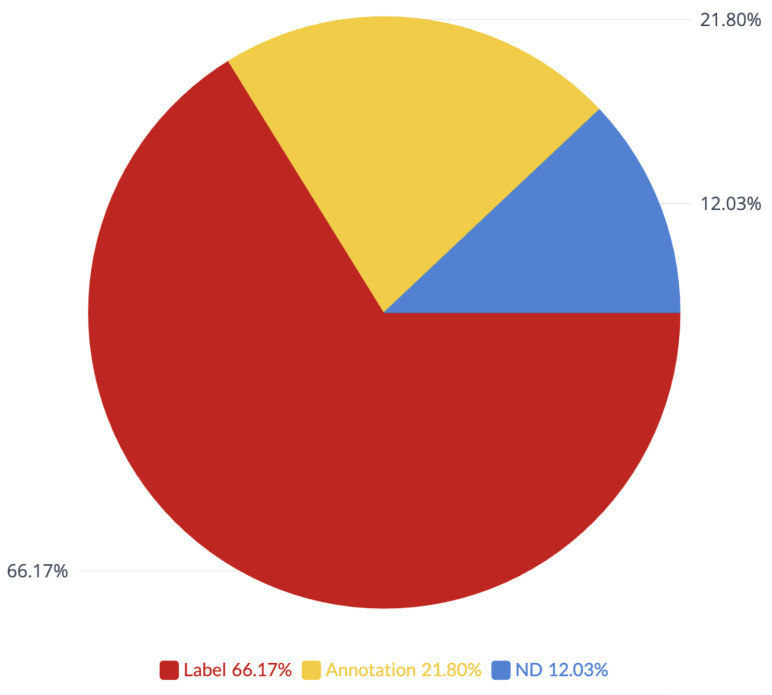
Breakdown of the image descriptions. Label = manually assigned labels corresponding to diagnosed ocular diseases, image quality or described areas of interest, Annotation = manual annotations on images indicating areas of interest, ND = no descriptions.

**Figure 5 jcm-12-03587-f005:**
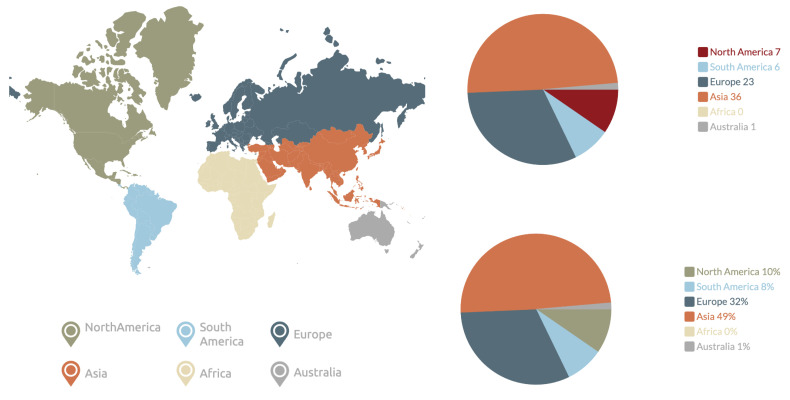
Geographical distribution of the available datasets by continent (where origins of the dataset could be determined).

**Figure 6 jcm-12-03587-f006:**
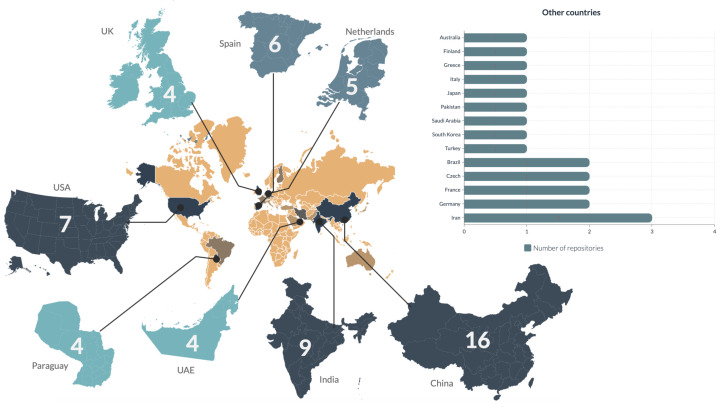
Number of datasets originating from individual countries (where origins of the dataset could be determined).

**Table 1 jcm-12-03587-t001:** Characteristics of the open access datasets.

ID	Name	Access		Technical Details						Epidemiological Details			Legality
		Access Type	Ease of Access	Number of Photos	Photo Sizes	Photo Descriptions	Size of the Repository	Additional Artifacts in the Photos	Repository URL	Number of Patients	Eye Diseases	Country of Origin	
1	1000FIWC	OAAR	2	1000	Different: 3000 × 3152 px; 1728 × 2592 px; …	Labels	777 MB	Absence	[12]	NR	DR, G	China	Citation: [128]
2	ACDLTFARIFP	OA	1	2206	350 × 346 × 3 px	Absence	294 MB	Absence	[13]	NR	NR	NR	Citation: [129]
3	ACRFAODAWDCMFERLRLASMAGD Version 1	OA	1	2560	Different: 456 × 951 px; 576 × 760 px; …	Labels	302 MB	Absence	[14]	NR	DME, CSR, AMD, G	UAE	Citation: [130]
4	ACRFAODAWDCMFERLRLASMAGD Version 2	OA	1	2988	Different: 456 × 951 px; 576 × 760 px; …	Labels	1597 MB	Absence	[15]	NR	G	UAE	Citation: [131]
5	ACRFAODAWDCMFERLRLASMAGD Version 3	OA	1	3100	Different: 1934 × 2032 px; 576 × 760 px; …	Labels	2969 MB	Absence	[16]	NR	DME, CSR, AMD, G	UAE	Citation: [132]
6	ACRFAODAWDCMFERLRLASMAGD Version 4	OA	1	2560	Different: 1934 × 2032 px; 576 × 760 px; …	Labels	1433.6 MB	Absence	[17]	NR	DME, CSR, AMD, G	UAE	Citation: [133,134]
7	ACRIMA	OA	3	705	Different: 349 × 349 px; 871 × 871 px; …	Labels	23 MB	Absence	[18]	NR	G	Spain	Citation: [135]
8	APTOS	OAAL	2	5590	Different: 640 × 480 px; 2416 × 1736; …	Labels	9748 MB	Absence	[19]	NR	DR	India	NR
9	Arteriovenous Nicking	OA	3	90	401 × 401 px	Labels	29 MB	Absence	[20]	NR	NR	NR	NR
10	CHHSIE	OA	3	28	999 × 960 px	Annotations	3 MB	Absence	[21]	14	HE	UK	NR
11	CTDRD	OAAR	2	35,136	Different: 4712 × 3163 px; 2213 × 2205 px; …	Absence	40,755 MB	Absence	[22]	NR	DR	NR	NR
12	Calibration level 1	OAAR	2	3200	Different: 1424 × 2144 px; 1536 × 2048 px;	Labels	8181 MB	Absence	[23]	NR	DR	India	Citation: [136]
13	Cardiacemboli	OAAR	2	943	Different: 3264 × 4928 px; 2048 × 3072 px; …	Labels	767 MB	Absence	[24]	NR	CE	NR	NR
14	DDR	OAAR	2	12524	Different: 512 × 512 px; 1536 × 2048 px; …	Labels	3348 MB	Absence	[25]	NR	DR	China	Citation: [137]
15	DEFRIOVAA	OA	3	50	Different: 798 × 658 px; 886 × 719 px; …	Absence	43.2 MB	Absence	[26]	50	DED	Netherlands	NR
16	DFFIFTSODR Version 0.1	OA	1	396	2124 × 2056 px	Labels	238 MB	Absence	[27]	NR	DR	Paraguay	Citation: [138]
17	DFFIFTSODR Version 0.2	OA	1	757	Different: 1444 × 1444 px; 2056 × 2124 px; …	Labels	375 MB	Absence	[28]	NR	DR	Paraguay	Citation: [138]
18	DFFIFTSODR Version 0.3	OA	1	1437	Different: 1028 × 1052 px; 1029 × 1062 px	Labels	1536 MB	Absence	[29]	NR	DR	Paraguay	Citation: [138]
19	DHRFID	OAAR	2	6542	Different: 3456 × 5184 px; 1904 × 2460 px; …	Labels	857 MB	Absence	[30]	NR	DR, G, O	NR	NR
20	DOFIFVSDOHRDRAP	OA	1	100	1504 × 1000 px	Absence	192 MB	Presence	[31]	NR	HR, DR, P	Pakistan	Citation: [139]
21	DOOAFI Version 1	OA	1	50	2032 × 1934 px	Absence	33 MB	Absence	[32]	NR	G	NR	Citation: [140]
22	DOOAFI Version 2	OA	1	50	2032 × 1 934 px	Absence	33 MB	Absence	[33]	NR	G	NR	Citation: [140]
23	DR (resized)	OAAR	2	70,234	Different: 278 × 278 px; 774 × 1024 px; …	Labels	8038 MB	Absence	[34]	NR	DR	China	NR
24	DR 1	OA	3	1077	640 × 480 px	Labels	200 MB	Absence	[35]	NR	DED	Brazil	Citation: [141]
25	DR 2	OA	3	520	Different: 873 × 500 px; 877 × 582 px; 876 × 581 px; …	Labels	200 MB	Absence	[36]	NR	DED	Brazil	Citation: [141]
26	DR 3	OAAR	2	13251	Different: 1184 × 1792 px; 2048, 3072 px; …	Labels	14,848 MB	Absence	[38]	NR	DR	NR	NR
27	DR 4	OAAR	2	105375	512 × 786 px	Labels	12,697 MB	Absence	[37]	NR	DR	NR	NR
28	DR 5	OAAR	2	3662	Different: 358 × 474 px; 1226 × 1844 px; …	Labels	8908 MB	Absence	[39]	NR	DR	NR	NR
29	DR 6	OAAR	2	301	2056 × 2124 px	Labels	128 MB	Absence	[40]	NR	DR	NR	NR
30	DR 7	OAAR	2	103	Different: 3264 × 4928 px; 2000 × 3008 px; …	Absence	101 MB	Absence	[41]	NR	NR	NR	NR
31	DR2015DCR	OAAR	2	35,126	224 × 224 px	Labels	1014 MB	Absence	[42]	NR	DR	NR	NR
32	DR224 × 2242019	OAAR	2	3662	224 × 224 px	Labels	238 MB	Absence	[43]	NR	DR	India	NR
33	DR224 × 224GF	OAAR	2	3662	224 × 224 px	Labels	426 MB	Absence	[44]	NR	DR	India	NR
34	DR224 × 224GI	OAAR	2	3662	224 × 224 px	Labels	157 MB	Absence	[45]	NR	DR	India	NR
35	DRA	OAAR	2	35,126	Different: 1024 × 1024 px; 779 × 1024 px; …	Labels	1024 MB	Absence	[46]	NR	DR	NR	NR
36	DRB	OAAR	2	49,703	512 × 512 px	Labels	2048 MB	Absence	[47]	NR	DR	NR	NR
37	DRBDC	OAAR	2	88,700	Different: 4752 × 3168 px; 2560 × 1920 px; …	Labels	98,017 MB	Absence	[48]	NR	DR	NR	NR
38	DRC	OAAR	2	70,234	Different: 278 × 278 px; 727 × 1024 px; …	Labels	8192 MB	Absence	[49]	NR	DR	NR	NR
39	DRC #2	OAAR	2	2608	224 × 224 px	Absence	33 MB	Absence	[50]	NR	NR	NR	NR
40	DRC 3	OAAR	2	2608	224 × 224 px	Absence	33 MB	Absence	[51]	NR	NR	NR	NR
41	DRD	OAAR	2	2750	256 × 256 px	Labels	350 MB	Absence	[52]	NR	DR	NR	NR
42	DRD 2	OAAR	2	2111	Different: 554 × 512 px; 424 × 512 px; …	Labels	269 MB	Absence	[53]	NR	DR	NR	NR
43	DRD 3	OAAR	2	3662	224 × 224 px	Labels	349 MB	Absence	[54]	NR	DR	NR	NR
44	DRD 4	OAAR	2	12,844	224 × 224 px	Labels	725 MB	Absence	[55]	NR	DR	NR	NR
45	DRHARMDAGI	OA	3	39	Different: 3216 × 2136 px; 2816 × 1880 px; …	Labels	10 MB	Presence	[56]	38	DED, HR, G, AMD	UK	Citation: [142]
46	DRIFONS	OA	3	110	600 × 400 px	Annotations	2.5 MB	Presence	[57]	55	HR	Spain	Citation: [143]
47	DRIFVS	OAAR	2	40	565 × 584 px	Annotations	30 MB	Absence	[58]	400	DED	Netherlands	NR
48	DRO	OAAR	2	35,128	Different: 278 × 278 px; 738 × 1024 px; …	Labels	6656 MB	Absence	[59]	NR	DR	NR	NR
49	DRPD	OAAR	2	13,970	128 × 128 px	Labels	46 MB	Absence	[60]	NR	DR	NR	NR
50	DRR300 × 300C	OAAR	2	16,798	Different: 300 × 300 px; 271 × 273 px; …	Labels	123 MB	Absence	[61]	NR	DR	NR	NR
51	DRS	OAAR	2	10024	Different: 315 × 400 px; 1957 × 2196 px; …	Absence	10,240 MB	Absence	[62]	NR	NR	NR	NR
52	DRTW	OAAR	2	37254	Different: 278 × 278 px; 899 × 1024 px; …	Labels	4915 MB	Absence	[63]	NR	DR	NR	NR
53	DRU	OAAR	2	34882	Different: 3456 × 5184 px; 1957 × 2196 px; …	Labels	32,768 MB	Absence	[64]	NR	DR	China	NR
54	DR_2000	OAAR	2	2000	Different: 3456 × 5184; 2056 × 3088 px; …	Labels	2048 MB	Absence	[65]	NR	DR	China	NR
55	DR_201010	OAAR	2	35,136	Different: 2560 × 1920 px; 2592 × 1944 px; …	Labels	38,809 MB	Absence	[66]	NR	DR	NR	NR
56	DeepDRiD	OA	1	320	Different: 1725 × 2230 px; 3072 × 3900; …	Absence	1464 MB	Absence	[67]	NR	NR	NR	Citation: [138]
57	DiaRetDB1 V2.1	OAAR	2	89	1152 × 1500 px	Annotations	137 MB	Absence	[68]	NR	DR	Finland	Citation: [144]
58	Diabetic	OAAR	2	2769	Different: 342 × 512 px; 434 × 512 px; …	Labels	353 MB	Absence	[69]	NR	DR	NR	NR
59	Diabetic Retinopathy Detection Processed	OAAR	2	51,500	Different: 224 × 224 px; 400 × 400 px; …	Labels	2764 MB	Absence	[70]	NR	DR	NR	NR
60	Drishti-GS	OAAR	2	101	Different: 1845 × 2050 px; 1763 × 2047 px; …	Annotations	341 MB	Absence	[71]	NR	NR	NR	NR
61	Drishti-GS1	OAAR	2	101	Different: 2048 × 1760 px; 2049 × 1749; …	Annotations, labels	350 MB	Absence	[72]	NR	G	India	Citation: [145,146]
62	EOptha Diabetic Retinopathy	OAAR	2	46	Different: 1696 × 2544 px; 1000 × 1504 px; …	Annotations	21 MB	Absence	[73]	NR	DR	NR	NR
63	EPACS	OA	2	88702	Different: 4752 × 3168 px; 4928 × 3264 px; …	Labels	84,203 MB	Absence	[74]	NR	DED	USA	NR
64	Eye Dataset Workshop	OAAR	2	12734	Different: 3264 × 3264 px; 2592 × 2592 px; …	Labels	672 MB	Absence	[75]	NR	DR	NR	NR
65	FID	OAAR	2	32	605 × 700 px	Absence	1 MB	Absence	[76]	NR	HE	NR	NR
66	FIRD	OA	3	268	2912 × 2912 px	Annotations	264 MB	Absence	[77]	39	NR	Greece	Citation: [147]
67	Fundus	OAAR	2	1600	Different: 1725 × 2230 px; 1727 × 2232 px; …	Labels	890 MB	Absence	[78]	NR	DR	NR	NR
68	Fundus Images	OAAR	2	650	Different: 2048 × 3072 px; 2048 × 3085	Labels	206 MB	Absence	[79]	NR	G	NR	NR
69	Fundus_DR	OAAR	2	61830	256 × 256 px	Labels	1710 MB	Absence	[80]	NR	DR	NR	NR
70	Fundusvessels	OAAL	1	3909	565 × 584 × 3 px	Absence	100 MB	Absence	[98]	NR	NR	China	Citation: [148]
71	GFDR 2	OAAR	2	3662	224 × 224 px	Labels	349 MB	Absence	[81]	NR	DR	NR	NR
72	Glaucoma Fundus	OA	3	1542	240 × 240 px	Labels	118.2 MB	Absence	[82]	1542	G	Republic of Korea	Citation
73	HEIME	OA	3	169	2196 × 1958 px	Annotations	300 MB	Absence	[83]	910	DED	USA	Citation: [149]
74	HRFQA	OA	3	45	3888 × 2592 px	Labels	68.1 MB	Absence	[84]	45	DED	Germany and Czech Republic	Citation: [150]
75	HRFQS	OA	3	36	3504 × 2336 px	Annotations, labels	73 MB	Absence	[84]	18	NR	Germany and Czech Republic	Citation: [151]
76	IARMD	OAAR	2	1200	2124 × 2056 px	Labels	606 MB	Absence	[85]	NR	AMD	China	NR
77	IDRID	OAAR	2	516	4288 × 2848 px	Labels	970 MB	Absence	[86]	NR	DED	India	Citation: [152]
78	INSFPIOTRAR	OAAF	2	40	2392 × 2048 px	Annotations	80 MB	Absence	[87]	NR	G	USA	Citation: [153]
79	INSFPIOTRS	OAAF	2	30	768 × 1019 px	Annotations, labels	80 MB	Presence	[87]	15	G	USA	Citation: [154]
80	IPM	OAAR	2	1200	2124 × 2056 px	Labels	608 MB	Absence	[88]	NR	M	China	Citation: [155]
81	IRC	OAAR	2	1032	Different: 2848 × 3408 px; 2848 × 3712 px; …	Labels	486 MB	Absence	[89]	NR	DR, DME	India	NR
82	ISBI_2021_Retina_23	OAAR	2	2560	Different: 1424 × 2144 px; 1536 × 2048 px; …	Labels	6400 MB	Absence	[90]	NR	O	NR	NR
83	ISBI_RETINA_TEST	OAAR	2	640	Different: 1424 × 2144 px; 1536 × 2048 px; …	Labels	1597 MB	Absence	[91]	NR	O	NR	NR
84	JSIEC	OA	3	1000	3046 × 2572 px	Labels	778 MB	Absence	[92]	NR	DED	China	Citation: [129]
85	Jichi DR	OA	3	9939	1272 × 1272 px	Labels	845 MB	Absence	[93]	2740	DED	Japan	NR
86	LSABG	OAAE	1	4854	500 × 500 px	Labels	300 MB	Absence	[94]	NR	G	China	Citation: [156]
87	Messidor-2	OAAF	2	1748	2240 × 1488 px	Absence	2355 MB	Absence	[95]	874	DED	France	Citation: [157]
88	ODIR	OAAR	1	8000	2048 × 1536	Labels	1228 MB	Absence	[96]	5000	DED, HR, G, AMD, C, M, O	China	NR
89	ODR	OAAR	2	14,392	Different: 188 × 250 px; 1607 × 2139 px; …	Labels	2078 MB	Absence	[97]	5000	D, G	China	NR
90	OFOAOS	OAAR	1	3909	740 × 740 × 3 px	Annotations	100 MB	Absence	[98]	NR	NR	China	NR
91	Papila Version 1	OA	1	490	2576 × 1934 px	Annotations, labels	563 MB	Absence	[99]	NR	M, H, A	Paraguay	Citation: [158]
92	Papila Version 2	OA	1	488	2576 × 1934 px	Annotations, labels	563 MB	Absence	[100]	NR	M, H, A	Spain	Citation: [158]
93	R20152019BDI	OAAR	2	94,292	Different: 149 × 1024 px; 825 × 1024 px; …	Labels	18,432 MB	Absence	[101]	NR	DR	Indie	NR
94	REFUGE Challenge 2020	OAAR	1	1200	Different: 2124 × 2056 px; 1634 × 1634 px	Annotations, labels	NR	Absence	[102]	NR	G	China	Citation: [159]
95	RFGC	OAAR	2	1200	1634 × 1634 px	Labels	1433 MB	Absence	[103]	NR	G	China	NR
96	RFI	OAAR	2	21,746	Different: 314 × 336 px; 753 × 1024 px; …	Labels	2048 MB	Absence	[104]	NR	DR, M, C	NR	NR
97	RFIFGA	OA	3	750	2376 × 1584 px	Annotations	13,209 MB	Absence	[105]	NR	G	Saudi Arabia, France	Citation: [160]
98	RFIR	OAAR	2	270	Different: 2912 × 2912 px; 2912 × 2912 px; …	Annotations	456 MB	Presence	[106]	NR	NR	NR	Citation: [147]
99	RIM-ONE Version 2	OA	3	455	Different: 398 × 401 px; 517 × 494 px; …	Annotations, labels	12 MB	Absence	[107]	NR	G	Spain	NR
100	RIM-ONE Version 3	OA	3	159	2144 × 1424 px	Annotations, labels	227 MB	Absence	[107]	NR	G	Spain	NR
101	RITE	OAAR	2	100	512 × 512 px	Annotations	33 MB	Absence	[108]	NR	NR	USA	Citation: [161]
102	RODRD	OAAF	3	1120	2000 × 1312 px	Labels	4710 MB	Absence	[109]	70	DED	Netherlands	Citation: [162]
103	RVCSF	OAAR	2	226	Different: 224 × 224 px; 605 × 700 px	Annotations	6 MB	Absence	[110]	NR	NR	NR	NR
104	Retina	OAAL	2	601	Different: 1848 × 1224 px; 2592 × 1728; …	Labels	3072 MB	Absence	[111]	NR	G, C, RD	NR	NR
105	Retina Online Challenge	OAAF	2	100	768 × 576 px	Annotations	26 MB	Absence	[112]	NR	DED	Netherlands	NR
106	Retina_Quality	OAAR	2	26,052	Different: 3246 × 3245 px; 2258 × 2257 px; …	Labels	3788 MB	Absence	[113]	NR	DR	NR	NR
107	Retinagen	OAAR	2	500	Different: 605 × 700 px; 1106 × 1280 px; …	Absence	23 MB	Absence	[114]	NR	NR	NR	NR
108	Retinal Vessel Tortuosity	OAAF	2	60	1200 × 900 px	Annotations	16 MB	Markers	[115]	34	HR	Italy	Citation, inform: [163]
109	Retinal_tiny	OAAR	2	2062	512 × 512	Labels	104 MB	Absence	[116]	NR	DR	NR	NR
110	SAOTR	OA	3	397	700 × 605 px	Annotations, labels	361 MB	Absence	[117]	NR	DED	USA	NR
111	SDRD	OAAR	2	1939	224 × 224 px	Labels	188 MB	Absence	[118]	NR	DR	NR	NR
112	SUSTech-SYSU	OAAR	2	1151	2136 × 2880 px	Labels	400 MB	Absence	[119]	NR	DR	China	Citation: [164]
113	Sydney Innovation Challenge 2019	OAAR	2	14,145	2448 × 3264 px	Labels	18,534 MB	Absence	[120]	NR	DR	Australia	NR
114	VACRDD	OAAR	2	3785	512 × 512 px	Labels	61 MB	Absence	[121]	NR	DR, G, O	NR	NR
115	VIFTCOTAVR	OAAE	1	56	768 × 576 px	Annotations	12 MB	Absence	[122]	NR	NR	Spain	Citation: [165]
116	WIDE	OA	3	30	Different: 731 × 1300 px; 977 × 1516 px; 854 × 1393 px…	Annotations	63 MB	Absence	[123]	30	AMD	USA	Citation: [166]
117	William Hoyt	OA	3	850	Different: 601 × 600 px; 596 × 600 px; …	Labels	170 MB	Presence	[124]	NR	P	NR	NR
118	Yangxi	OA	3	18,394	297 × 297px	Labels	5529 MB	Absence	[125]	5825	AMD	China	NR
119	dr15_test	OAAR	2	34,043	512 × 512 px	Labels	10,240 MB	Absence	[126]	NR	DR	NR	NR
120	merged_retina_datasets	OAAR	2	2451	Different: 2847 × 3925 px; 2847 × 3414 px; …	Labels	5816 MB	Absence	[127]	NR	DR, O	NR	NR

**Dataset acronyms**: APTOS = Asia Pacific Tele-Ophthalmology Society, CHHSIE = Child Heart Health Study in England, DEFRIOVAA = Digital Extraction from Retinal Images of Veins and Arteries, DRHARMDAGI = Diabetic Retinopathy Hypertension, Age-Related Macular Degeneration and Glaucoma Images, DRIFONS = Digital Retinal Images for Optic Nerve Segmentation, DRIFVS = Digital Retinal Images for Vessel Segmentation, EPACS = Eye Picture Archive Communication System, FIRD = Fundus Image Registration Dataset, HEIME = Hamilton Eye Institute Macular Edema, HRFQS = High-Resolution Fundus Quality Segmentation, HRFQA = High-Resolution Fundus Quality Assessment, IARMD = iChallenge Age-Related Macular Degeneration, IPM = iChallenge Pathological Myopia, IDRID = Indian Diabetic Retinopathy Image Dataset, INSFPIOTRAR = Iowa Normative Set for Processing Images of the Retina—Arteriovenous Ratio, INSFPIOTRS = Iowa Normative Set for Processing Images of the Retina—Stereo, JSIEC = Joint Shantou International Eye Center, LSABG = Large-Scale Attention-based Glaucoma, RODRD = Rotterdam Ophthalmic Data Repository DR, ODIR = Ocular Disease Intelligent Recognition, RFGC = Retina Fundus Glaucoma Challenge, RFIFGA = Retinal Fundus Images for Glaucoma Analysis, SAOTR = Structured Analysis of the Retina, VIFTCOTAVR = VARPA Images for the Computation of the Arterio/Venular Ratio, DOFIFVSDOHRDRAP = Data on Fundus Images for Vessel Segmentation, Detection of Hypertensive Retinopathy, Diabetic Retinopathy and Papilledema, DFFIFTSODR Version 0.1 = Dataset from Fundus Images for the Study of Diabetic Retinopathy Version 0.1, DFFIFTSODR Version 0.2 = Dataset From Fundus Images for the Study of Diabetic Retinopathy Version 0.2, DFFIFTSODR Version 0.3 = Dataset from Fundus Images for the Study of Diabetic Retinopathy Version 0.3, ACRFAODAWDCMFERLRLASMAGD Version 1 = A Composite Retinal Fundus and OCT Dataset along with Detailed Clinical Markings for Extracting Retinal Layers, Retinal Lesions and Screening Macular and Glaucomatous Disorders Version 1, ACRFAODAWDCMFERLRLASMAGD Version 2 = A Composite Retinal Fundus and OCT Dataset along with Detailed Clinical Markings for Extracting Retinal Layers, Retinal Lesions and Screening Macular and Glaucomatous Disorders Version 2, ACRFAODAWDCMFERLRLASMAGD Version 3 = A Composite Retinal Fundus and OCT Dataset along with Detailed Clinical Markings for Extracting Retinal Layers, Retinal Lesions and Screening Macular and Glaucomatous Disorders Version 3, ACRFAODAWDCMFERLRLASMAGD Version 4 = A Composite Retinal Fundus and OCT Dataset along with Detailed Clinical Markings for Extracting Retinal Layers, Retinal Lesions and Screening Macular and Glaucomatous Disorders Version 4, ACDLTFARIFP = A CycleGAN Deep Learning Technique for Artifact Reduction in Fundus Photography, OFOAOS = ORIGA for OD and OC Segmentation, DeepDRiD = Deep-Diabetic- Retinopathy- Image-Dataset- (DeepDRiD), DOOAFI Version 1 = Data on OCT and Fundus Images Version 1, DOOAFI Version 2 = Data on OCT and Fundus Images Version 2, DR224 × 224GF = Diabetic Retinopathy 224 × 224 Gaussian Filtered, DR2015DCR = Diabetic Retinopathy 2015 Data Colored Resized, DRA = Diabetic Retinopathy Arranged, RFIR = Retina Fundus Image Registration, DR224 × 2,242,019 = Diabetic Retinopathy 224 × 224 (2019 Data), DR224 × 224GI = Diabetic Retinopathy 224 × 224 Grayscale Images, Drishti-GS = Drishti-GS—RETINA DATASET FOR ONH SEGMENTATIO N, DRU = Diabetic Retinopathy Unziped, RVCSF = Retinal Vessel Combine Same Format, DRR300 × 300C = Diabetic-Retinopathy-Resized-300 × 300-Cropped, VACRDD = VietAI Advance Course—Retinal Disease Detection, 1000FIWC = 1000 Fundus Images with 39 Categories, DR (resized) = Diabetic Retinopathy (resized), DHRFID = Derbi_Hackathon _Retinal_Fundus _Image_Dataset, SDRD = Small_Diabetic_ Retinopathy_Dataset, DRD = Diabetic Retinopathy Dataset, R20152019BDI = Resized 2015 & 2019 Blindness Detection Images, DRPD = Diabetic Retinopathy Preprocessed Dataset, DRC = Diabetic Retinopathy Classified, DRD 2 = Diabetic Retinopathy Detection, DRB = Diabetic_Retinopathy_Balanced, DRD 3 = Diabetic Retinopathy Detection, DRBDC = Diabetic Retinopathy Blindness Detection c_data, DRO = Diabetic Retinopathy Organized, Diabetic Retinopathy Detection Processed = Diabetic Retinopathy Detection Processed, CTDRD = Cropped-Train-Diabetic-Retinopathy-Detection, DRTW = Diabetic-Retinopathy-Train- Validation, GFDR 2 = Gaussian_Filtered _Diabetic_Retinopathy, DRC #2 = Diabetic Retinopathy Classification #2, DRC 3 = Diabetic Retinopathy Classification, Sydney Innovation Challenge 2019 = Sydney Innovation Challenge 2019. Access type: OA = Open access, OAAR = Open access after registration, OAAE = open access after sending email, OAAR = open access after login, OAJC = open access after joining the contest and acceptance of the registration, NR= not reported. Diseases: G = glaucoma, HE = healthy eyes, DR = diabetic retinopathy, C = cataracts, RD = retinal diseases, DED = diabetic eye disease, HR = hypertensive retinopathy, AMD = age-related macular degeneration, M = myopia, O = other diseases, P = papilledema, H = hyperopia, A = astigmatism, CSR = central serous retinopathy, DME = diabetic macular edema, CE = cardiac embolism, D = diabetes, RVO = retinal vascular occlusion.

## Data Availability

Data sharing not applicable No new data were created or analyzed in this study. Data sharing is not applicable to this article.

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
