# Peer review of "A Global Review of Publicly Available Datasets Containing Fundus Images: Characteristics, Barriers to Access, Usability, and Generalizability"

_jcm, 2023, doi:10.3390/jcm12103587_

Round 1

Reviewer 1 Report

Thank you for the opportunity to review this manuscript that is very well written. The topic of interest--- publicly available fundus color photos datasets--- is highly relevant and important especially for the field of deep learning. Here are my minor comments:

1. More information in the introduction will be helpful. Specifically, the way that the authors categorized the datasets, as illustrated in Figures 1 to 5, was great. Please provide a preview in the introduction for the readers on how you plan to categorize the datasets--- essentially provide a high level overview of Figures 1 to 5.

2. The authors used the term "fundus images" throughout the manuscript. I assume they meant color fundus photos. Please be more specific and change the wording accordingly. For example, technically an autoflorescence image is a fundus image.

3.   “9 datasets had additional artifacts.” What do you mean by artefacts? Please specify

4. The comparison to Khan's article (reference 11) is critical. Please expand. Specifically, please indicate how many datasets with color fundus photos were included in reference 11, and specify what exactly happened to those datasets at the time of review for this particular article. E.g. Out of the XXX datasets with color fundus photos included in Khan's article, YYY were still available.

5. Can remove "file type" column in Table 2, as all of them are color fundus photos.

6. Please expand more on "generator of pseudo-random samples." What's the point of this generator? What is the goal? Provide a flow chat on how exactly this generator works.

Reviewer 2 Report

The authors summarize available image datasets. I think it will be a valuable index for the available dataset.

The authors summarized the latest publicly available datasets containing fundus images. The cited paper number 11 already summarized the same datasets two years ago;  therefore, I think there are few originalities in the field. The present study added the available datasets in these two years on the cited paper number 11. The methodology that the authors used was good. The conclusions are consistent with the evidence and arguments presented and they address the main question posed.  The references are appropriate.

The figures posted are small and hard to see, and the color combination is quite difficult for people with color blindness to distinguish.

Reviewer 3 Report

Thank you for submitting your manuscript. There are a couple of points that need to be addressed.

1- There is a lot of redundancy in the abstract and manuscript, which needs scientific edits.

2- Usually ''Method'' section is mentioned after ''Introduction''.

3- How did you ensure the comprehensibility of your search with the current key terms?

4- The way you organized the data in the tables makes it hard to read.

Reviewer 4 Report

The article provides an overview of the image repository for ophthalmic imagery.

This is very useful work for the optics community, and the authors did a commendable job of reviewing the literature and summarizing the findings.

The main issue with the paper is that it does not provide quick access to the information, and it is quite verbose.

The vast majority of people reading this work are looking for a clear table describing 

1. Type of images

2. Statistics of the population and possible demographic

3. Link to access the data or quick method to get it

4. List of papers connected to the data set.

The various pie chart (fig 1 2 and 3 and 4) are for most researchers not very useful.

The author should revise their submission in consideration of their audience.

Round 2

Reviewer 4 Report

My concerns have been addressed. Thank you.